# Playing games with multiple access channels

Felix Leditzky [1,2 ✉], Mohammad A. Alhejji[1,3], Joshua Levin[1,3] & Graeme Smith[1,2,3]

Communication networks have multiple users, each sending and receiving messages. A multiple access channel (MAC) models multiple senders transmitting to a single receiver, such as the uplink from many mobile phones to a single base station. The optimal performance of a MAC is quantified by a capacity region of simultaneously achievable communication rates. We study the two-sender classical MAC, the simplest and best-understood network, and find a surprising richness in both a classical and quantum context. First, we find that quantum entanglement shared between senders can substantially boost the capacity of a classical MAC. Second, we find that optimal performance of a MAC with bounded-size inputs may require unbounded amounts of entanglement. Third, determining whether a perfect communication rate is achievable using finite-dimensional entanglement is undecidable. Finally, we show that evaluating the capacity region of a two-sender classical MAC is in fact NP-hard.

[1] JILA, University of Colorado/NIST, Boulder, CO 80309, USA. [2] Center for Theory of Quantum Matter, University of Colorado, Boulder, CO 80309, USA. [3] Department of Physics, University of Colorado, Boulder, CO 80309, USA. ✉email: felix.leditzky@jila.colorado.edu

nformation theory is the mathematical theory of communication and signal processing pioneered by Shannon[1]. In network communication settings, the simplest model is a multiple access channel (MAC), where two spatially separated senders aim to transmit individual messages to a single receiver. Faithful information transmission through a MAC is possible within its capacity region, which was characterized by Ahlswede[2] and Liao[3] in terms of a so-called single-letter formula, i.e., an entropic optimization problem of fixed bounded dimension that is in principle computable. In quantum information theory, communication tasks can be enhanced dramatically if the communicating parties are given access to quantum resources such as shared entanglement[4,5]. However, certain tasks such as classical single-sender-single-receiver communication receive no advantage from entanglement assistance[6].

In this work, we show that MACs behave in a fundamentally different way in the presence of entanglement assistance, in contrast to the single-sender-single-receiver scenario. Moreover, even unassisted classical MACs exhibit far more complex behavior than previously widely appreciated. We demonstrate this by constructing a family of classical MACs with surprisingly rich behavior: First, we show that entanglement shared between the senders can strictly increase the capacity region of a classical MAC, proving that entanglement can help in a purely classical communication scenario. Second, we exhibit examples of channels for which an unbounded amount of entanglement is needed to achieve the maximal possible increase of the achievable rate region. We also show that it is generally undecidable to determine whether the maximal rate pair can be achieved for a MAC with finite-dimensional entanglement strategies. Finally, we prove in the unassisted communication setting that it is NP-hard to determine which rates can be achieved for a given MAC. Our findings imply that even for the arguably simplest network information-theoretic setting of a MAC, there is no general solution to the problem of efficiently determining its unassisted communication capabilities, highlighting the need for practical approximation algorithms. At the same time, entanglement assistance can push the achievable information transmission rates of MACs beyond the classical limit, paving the way for harnessing entangled resources in hybrid classical-quantum information networks.

## Results

**Entanglement helps a classical multiple access channel.** We first briefly review classical MAC, a general example of which is shown in Fig. 1. Our results concern the capacity region $\mathcal{C}(N)$ of a MAC

$N$, consisting of all the rate pairs $(R_1, R_2)$ such that sender $i$ can faithfully transmit information to the receiver at the rate $R_i$ (see Section 1.2 in the Supplementary Information for a more detailed definition). Ahlswede[2] and Liao[3] proved that $\mathcal{C}(N)$ is given by the convex hull of all pairs $(R_1, R_2)$ satisfying

$$R_1 \leq I(A; Z|B) \quad R_2 \leq I(B; Z|A) \quad R_1 + R_2 \leq I(AB; Z), \qquad (1)$$

for some product distribution $\pi_A \pi_B$ on $\mathcal{A} \times \mathcal{B}$. Here, $I(U; V|W) = H(UW) + H(VW) - H(W) - H(UVW)$ is the conditional mutual information, $H(X) = -\sum_i p(x_i) \log p(x_i)$ is the Shannon entropy of a random variable $X \sim p$ with the logarithm taken to base 2, and $I(U; V) = H(U) + H(V) - H(UV)$ is the mutual information.

The central object in our work is a multiple access channel $N_G$ defined in terms of a nonlocal game $G = (\mathcal{X}_1, \mathcal{X}_2, \mathcal{Y}_1, \mathcal{Y}_2, W)$, where $\mathcal{X}_1, \mathcal{X}_2$ and $\mathcal{Y}_1, \mathcal{Y}_2$ are the question and answer sets for Alice and Bob, respectively, and $W \subset \mathcal{X}_1 \times \mathcal{X}_2 \times \mathcal{Y}_1 \times \mathcal{Y}_2$ is the winning condition. For the MAC $N_G$, the input alphabets of the two senders Alice and Bob are the question-answer sets $\mathcal{X}_1 \times \mathcal{Y}_1$ and $\mathcal{X}_2 \times \mathcal{Y}_2$, respectively, and the output alphabet of $N_G$ is $\mathcal{X}_1 \times \mathcal{X}_2$. If Alice and Bob win the nonlocal game $G$, that is, $(x_1, y_1, x_2, y_2) \in W$, the channel is noiseless and outputs the question pair $(x_1, x_2)$ to the receiver. If they lose the game, $(x_1, y_1, x_2, y_2) \notin W$, the channel outputs a question pair $(\hat{x}_1, \hat{x}_2)$ drawn uniformly at random from $\mathcal{X}_1 \times \mathcal{X}_2$. More formally, the MAC $N_G : (\mathcal{X}_1 \times \mathcal{Y}_1) \times (\mathcal{X}_2 \times \mathcal{Y}_2) \rightarrow \mathcal{X}_1 \times \mathcal{X}_2$ is defined as

$$N_G(\hat{x}_1, \hat{x}_2 | x_1, y_1; x_2, y_2) := \begin{cases} \delta_{x_1 \hat{x}_1} \delta_{x_2 \hat{x}_2} & \text{if } (x_1, x_2, y_1, y_2) \in W, \\ (|\mathcal{X}_1||\mathcal{X}_2|)^{-1} & \text{else}. \end{cases}$$

$$(2)$$

This channel construction is inspired by previous work by Quek and Shor[7], who used a similar construction in terms of the CHSH game[8] for an interference channel consisting of two senders and two receivers. It also appeared in unpublished work by Nötzel and Winter (A. Winter, personal communication), a portion of which has appeared in ref. [9].

The noise in the MAC $N_G$ defined in (2) is determined by the players' ability to win the nonlocal game $G$. Clearly, if there exists a perfect strategy for Alice and Bob (i.e., a strategy that wins the game with certainty on any question pair), they can select their questions uniformly at random and transmit information to the receiver at rates $R_i = \log |\mathcal{X}_i|$, achieving the maximal possible sum rate $R_1 + R_2 = \log |\mathcal{X}_1| + \log |\mathcal{X}_2|$. On the other hand, if they cannot win the game with certainty, then the channel necessarily adds noise to their signals, and consequently the achievable sum rate decreases. We can make this intuition precise

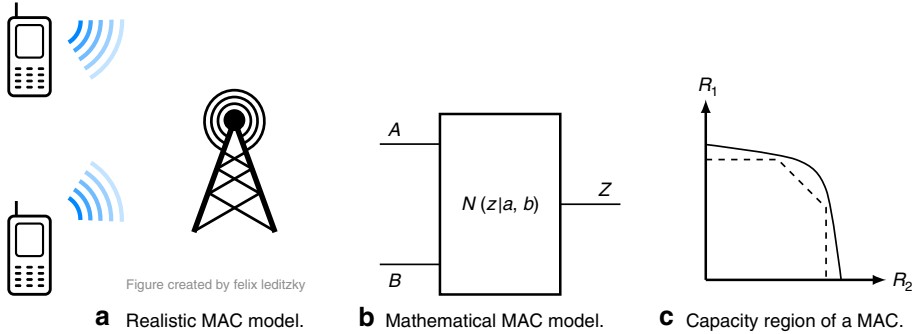

**Fig. 1 Multiple access channels. a** Realistic scenario of a multiple access channel (MAC), in which two cell phones send data to a cell tower. **b** Mathematical model of a MAC $N$, characterized by finite input alphabets $\mathcal{A}$ and $\mathcal{B}$, an output alphabet $\mathcal{Z}$, and a conditional probability distribution $N(z|a, b)$ for $a \in \mathcal{A}, b \in \mathcal{B}, z \in \mathcal{Z}$. The random variables corresponding to the senders and the receiver are denoted by $A$, $B$, and $Z$, respectively. **c** A typical capacity region of a MAC (solid line), together with an achievable pentagonal region for a fixed input distribution (dashed lines).

by observing that, setting $A = X_1 Y_1$ and $B = X_2 Y_2$, the mutual information $I(X_1 Y_1 X_2 Y_2; Z)$ constraining the sum rate $R_1 + R_2$ in (1) can be expressed as:

$$I(X_1 Y_1 X_2 Y_2; Z) = H(Z) - p_L (\log |\mathcal{X}_1| + \log |\mathcal{X}_2|). \quad (3)$$

Here, $p_L$ denotes the probability of losing the game $G$ given a product distribution $p_A p_B$ on the questions $x_i$ and a strategy producing the answers $y_i$. This relation allows us to prove a bound on the capacity region of $N_G$ whenever the nonlocal game $G$ does not admit a perfect strategy:

**Theorem 1** *If a nonlocal game $G$ does not admit a perfect strategy (using the available resources), the sum rate $R_1 + R_2$ of the MAC $N_G$ defined in (2) is strictly bounded away from $\log |\mathcal{X}_1| + \log |\mathcal{X}_2|$.*

Consider now a nonlocal game $G$ that cannot be won with certainty using any classical strategy, and assume there exists a perfect quantum strategy. If we allow shared entanglement between the two senders of the MAC, then Theorem 1 provides a provable separation between the capacity region of the unassisted MAC $N_G$ and the entanglement-assisted achievable rate region. A well-known example of such a nonlocal game is the magic square game $G_{MS}$[10–13], in which Alice and Bob have to fill a given row and column of a $3 \times 3$-square with a bit string such that the parity of Alice's row is even, the parity of Bob's column is odd, and the assignments are consistent in the overlapping cell. It is well-known that any classical strategy has winning probability at most $\frac{8}{9}$[13], as illustrated in the left panel in Fig. 2. For the MAC $N_{G_{MS}}$ defined in terms of the magic square game, we can use Theorem 1 to obtain an upper bound on the achievable sum rate of 3.13694, bounding it away from the maximal value of $2\log 3 \approx 3.17$.

On the other hand, there is a perfect quantum strategy in which Alice and Bob make measurements on two maximally entangled states shared between them[10,11,13]. This perfect quantum strategy, depicted in the right panel of Fig. 2, allows Alice and Bob to send individual messages to the receiver at the rates $R_1 = R_2 = \log 3$, yielding the maximal sum rate $R_1 + R_2 = 2\log 3$, which is not achievable by any classical coding strategy by Theorem 1. Therefore, the unassisted capacity region of $N_{G_{MS}}$ is separated from the point $(\log 3, \log 3)$, while the entanglement-assisted achievable rate region includes this point. This separation between classical strategies and entanglement-assisted strategies occurs for any nonlocal game $G$ with no perfect classical strategies and perfect quantum strategies, a so-called "pseudo-telepathy game"[13]. Each game in this class yields a separation for the corresponding MAC between the unassisted capacity region and the entanglement-assisted region via Theorem 1.

**How much entanglement do you need?**. Our main result, Theorem 1, can also be applied to the separate achievable rate regions for coding strategies using different amounts of entanglement. To illustrate this, we consider the class of linear system games $G_{LS}$[14], which are defined in terms of an $m \times n$ linear system $Ax = b$ of equations over $\mathbb{F}_2$. Slofstra and Vidick showed that there is a particular instance $G_{SV}$ of a linear system game for which a perfect winning strategy is necessarily quantum and furthermore requires an unbounded amount of entanglement[15]. More precisely, they gave upper and lower bounds on the local dimension $d$ of the quantum systems associated with Alice and Bob in the quantum strategy in terms of the losing probability $p_L$. Consider now the MAC $N_{G_{SV}}$ defined according to (2) in terms of the linear system game $G_{SV}$. Limiting Alice and Bob to entanglement assistance of local dimension at most $d$, their probability of losing the linear system game is strictly positive[15]. Consequently, we can invoke Theorem 1 to conclude that the $d$-dimensional entanglement-assisted achievable rate region of $N_{G_{SV}}$ is bounded away from the rate pair $(\log m, \log n)$ achieving the maximal sum rate $\log m + \log n$. On the other hand, it is straightforward to define a $d$-dimensional entanglement-assisted coding strategy for Alice and Bob based on the quantum strategy derived by Slofstra and Vidick[15] whose winning probability converges to unity as $d$ grows. Hence, as Alice and Bob have access to larger and larger entangled states, they approximate the perfect sum rate $\log m + \log n$ arbitrarily well.

Our results show that linear system games give rise to a family of MACs whose $d$-entanglement-assisted achievable rate regions approach the rate pair $(\log m, \log n)$ in the limit $d \to \infty$, yet they are strictly bounded away from it for any fixed finite $d$. Moreover, considering all finite-dimensional quantum strategies for a general linear system game $G_{LS}$, Slofstra showed that it is undecidable to determine whether there is a perfect quantum strategy among them[16]. By the arguments above, this directly translates to the following result: For the MAC $N_{G_{LS}}$ defined in terms of a linear system game $G_{LS}$, it is undecidable to determine whether the entanglement-assisted achievable rate region includes the rate pair $(\log m, \log n)$.

**Complexity of the capacity region of a classical MAC**. Finally, we turn our focus to the unassisted coding scenario for a discrete MAC. In information-theoretic terms, this scenario seems well understood as the capacity region $\mathcal{C}(N)$ of a MAC $N$ can be expressed in terms of a computable single-letter formula[2,3]. However, the single-letter nature of the capacity region formula by itself does not guarantee an efficient method of computing $\mathcal{C}$ in, say, runtime polynomial in $\max\{|\mathcal{A}|, |\mathcal{B}|, |\mathcal{Z}|\}$, the maximal size of the input and output alphabets of $N$. Using our construction of a MAC in terms of nonlocal games, we prove that it is NP-hard to decide whether a given point $(R_1, R_2)$ belongs to the capacity region of a MAC to within an additive error inverse-cubic in $n$, where $n$ is the size of the output alphabet. Our result is based on a nonlocal game version $G_H$ of 3SAT introduced by Håstad[17], consisting of $m = \mathcal{O}(n)$ clauses containing exactly three out of $n$ literals. For this nonlocal game, it follows from the probabilistically checkable proofs theorem[18,19] that it is NP-hard to decide if there is a perfect winning strategy for $G_H$ or if the maximal winning probability is bounded from above by $1 - (1 - c)/n$ for some constant $c < 1$[17]. Consider now $N_{G_H}$, the MAC defined in (2) in terms of the game $G_H$. Clearly, if Alice and Bob have a perfect winning strategy for $G_H$, they can each code at the rates $R_1 = \log m$ and $R_2 = \log n$ by choosing a uniform distribution over their respective question alphabets. This leads to the maximal sum rate $R_1 + R_2 = \log m + \log n$. On the other hand, if the maximal winning probability is bounded from above by $\omega^* = 1 - (1 - c)/n$, then Theorem 1 can be used to show that the sum rate $R_1 + R_2$ is

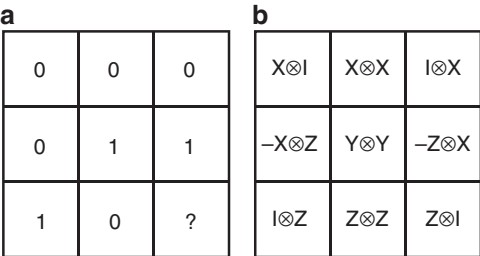

**a**

| 0 | 0 | 0 |
|---|---|---|
| 0 | 1 | 1 |
| 1 | 0 | ? |

**b**

| $X \otimes I$ | $X \otimes X$ | $I \otimes X$ |
|---|---|---|
| $-X \otimes Z$ | $Y \otimes Y$ | $-Z \otimes X$ |
| $I \otimes Z$ | $Z \otimes Z$ | $Z \otimes I$ |

**Fig. 2 Classical and quantum strategies for the magic square game. a** An optimal classical strategy for the magic square game that allows Alice and Bob to win the game for 8 of the 9 possible questions. Filling the bottom right square consistently with the parity constraints for the rows (even) and columns (odd) is impossible. **b** A perfect quantum strategy defined by measuring the observables in the cells on two maximally entangled states. Note that the observables along each row and column commute pairwise.

bounded from above by $\log m + \log n - (1 - \omega^*)^3$. In this case, the capacity region $\mathcal{C}(N_{G_H})$ is bounded away from the rate pair $(\log m, \log n)$. Altogether, this shows that it is NP-hard to decide if an arbitrary rate tuple $(R_1, R_2)$ belongs to $\mathcal{C}(N_{G_H})$ to precision inverse-cubic in $n$.

## Discussion

In this work we show that the capacity region of a multiple access channel displays complex behavior, both in a purely classical setting and when the senders have access to shared entangled quantum states. In particular, we prove that entanglement assistance can boost the achievable rates in a setting where two senders try to convey classical information through a common classical communication channel to a single receiver. Such an increase in capacity is impossible in the point-to-point scenario involving a single sender and a receiver. We also show that for a certain family of MACs the two senders need to share an unbounded amount of entanglement in order to achieve the ideal communication rate pair. When restricted to finite-dimensional entangled strategies, it is undecidable for this particular channel family whether the ideal rate can be achieved. Finally, we show that even in the unassisted scenario, it is in fact NP-hard to decide whether the ideal rate pair belongs to the capacity region of a MAC. This result is a strong counterpoint to the widely held belief that the availability of a computable single-letter formula for the capacity region essentially solves the MAC problem. The central tool in the proofs of all results above is the construction of a MAC in terms of a nonlocal game in such a way that the noise level of the channel is determined by the senders' ability to win the game.

Our work opens up a number of interesting topics for future work. Numerical investigations for the magic square channel suggest that the true separation between classical and quantum coding strategies for the MACs considered in this work is considerably larger than the separation guaranteed by Theorem 1, suggesting that our bound on the sum rate could be further tightened. For our results above we considered a specific achievable rate region that arises naturally when the two senders measure identical copies of a single entangled state. In general, the senders might have access to multipartite entangled states and implement parallel encoding strategies, which leads to the notion of an entanglement-assisted capacity region. We expect this region to be given by the regularization of the achievable region considered in this paper. For the MACs defined via our construction, this question seems to be related to parallel repetition theorems for nonlocal games played with quantum strategies. Furthermore, in this work we only considered entanglement shared between the two senders, and the communication setting could be generalized to one where entanglement is shared between both the senders and the receiver. Finally, our NP-hardness result for the unassisted capacity region of a MAC underlines the need for tight efficiently computable outer bounds on the unassisted capacity region. Such bounds could for example be obtained from convex relaxations of the rank-1 optimization problem describing the MAC capacity region[20].

## Data availability

No data sets were generated during this study.

## Code availability

MATLAB and Mathematica code files used to obtain the numerical bounds in Supplementary Note 3 are available from the corresponding author upon request.

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

## Acknowledgements

We thank Mark M. Wilde and Andreas Winter for helpful comments and feedback. This work was partially supported by NSF Grant No. PHY 1734006 and NSF CAREER award CCF 1652560.

## Author contributions

F.L., M.A., J.L., and G.S. each contributed extensively to the paper.

## Competing interests

The authors declare no competing interests.
