## [Peer Review File · Nature Communications]

REVIEWERS' COMMENTS:

Reviewer #1 (Remarks to the Author):

The multiple access channel MAC is one of the few network information theory problems for which the capacity region is understood, in the sense that there exists a single-letter formula characterising the achievable rate pairs. This paper shows several results suggesting that MAC channels exhibit a behaviour that can be much more complicated than point to point channels

* The main result of the paper is to show that one can construct completely classical multiple access channels for which entanglement between the senders increases the capacity region. Note that this is surprising because it is well known that in the point to point case, entanglement does not increase the capacity of classical channels (even if it can have a small effect in the one-shot setting). The way the authors prove this result is by constructing a MAC whose inputs correspond to input/output pairs of a non-local game, and the output is the two inputs if the game is won and uniformly random otherwise. Then, by taking a game with a perfect entangled strategy and no perfect classical strategy, they show that with a quantum strategy, one can obtain maximal rates, but not with a classical strategy.

* Another very interesting result that follows from the connection to non-local games is the NP-hardness of deciding whether a point is in the rate region of a given channel. This result is also surprising as MAC channels are usually considered "easy" in the information theory literature.

* Using recent advances in the understanding of non-local games, the authors show that unbounded entanglement might be needed to achieve the full entanglement-assisted rate region.

I think this paper gives a new perspective on the MAC channel, which is one of the fundamental channels in network information theory, and shows that it might not be as simple as it appeared. For all these reasons, I recommend acceptance of this paper.

Omar Fawzi

Reviewer #2 (Remarks to the Author):

This is a succinct and well-written paper. The author's have three key findings that should be of interest to many in this community. They show that when quantum entanglement is used as a resource, that it can substantially boost the capacity of a multiple access channel(MAC). This is not surprising but the way that they show this fact is novel. Next they show that there are cases where one cannot achieve the optimal entanglement-assisted performance of a MAC with a finite amount of entanglement. Their third key result is that determining if the optimal entanglement-assisted performance is achievable with a finite amount of entanglement is undecidable.

Their main construction, which allows them to obtain these striking results, is a clever construction of a multiple access channel from a non-local game. This construction then allows them to import recent results of Slofstra and Vidick from the theory of non-local games into the theory of MAC's.

I have verified the mathematical claims made in the appendices, so the paper should be accepted.

Reviewer #3 (Remarks to the Author):

Based on encoding non-local games in (classical) MACs, the authors derive four results:

- (1) Quantum assistance can boost the capacity of MACs.
- (2) The dimension of the quantum assistance required for achieving the quantum assisted capacity is in general unbounded (even for MACs of fixed dimension).
- (3) Determining if the quantum assisted capacity is achievable with finite-dimensional quantum assistance is undecidable.
- (4) The computation of the plain capacity of MACs is NP-hard.

As far as I know, all of these findings are novel. All the proofs are correct. The paper is well-written which makes the results accessible to a broad audience. The methods sections are of high quality as well. I particularly like the following aspects of the submission:

- There is the fundamental question on how quantum correlations influence network Shannon information theory and the submission makes an important contribution in that regard. We knew from previous work that for the single sender-receiver setting Shannon theory basically stays the same when adding quantum correlations (or even more general non-signalling correlations). The authors now show that already for the simple MAC setting this is no longer the case and capacity regions are improved by quantum correlations. This is an exciting and conceptually rich contribution. (I should add that there is also the previous work [7] about interference channels but I consider the work here to be more elegant and conceptually more clear.)
- I expect that the connection made between non-local games and channel coding to lead the way to plentiful strong follow-up works.

There are some weaknesses as well:

- The submission is a collection of results, where I find the result (1) to be more interesting than the others. Results (2)-(3) are nice to have but basically follow immediately from the known literature on non-local games (once the connection of channel coding to non-local games is made). Result (4) is independent but may not be that groundbreaking. I would argue that the NP-hardness is not that surprising and moreover pretty straightforward to get from known complexity-theoretic results.
- One might ask what the actual capacity is for general MACs with quantum assistance and this question is left open (same question for non-signalling assistance). However, this could very well be a hard question with no immediate answer and one might then also argue that this is where the follow-up works will come in.

Overall, I am positive about the submission and I recommend publication in Nature Communications (up to my minor comments below). Novel results in network Shannon theory are hard to derive by now and I judge that the submission will generate broad interest in the classical and quantum information theory community and beyond.

Comments:

- In line 22 (also line 120), the term "so-called single-letter formula that is in principle computable" might not be very clear to a general reader. Maybe one could refer to it as "an entropic optimisation problem of fixed, bounded dimension" or something along those lines.
- In lines 27-28 (also lines 146-151), I am not sure that I agree with "even unassisted classical MACs exhibit far more complex behavior than previously appreciated". After all, the MAC formula is an optimisation over product inputs which we know from optimisation theory to generally be hard. Also there are previous papers on the subject (e.g., [19]) and as far as I can tell it was actually appreciated that the computation of the MAC region can be hard (of course in the submission under review this is now made precise in some complexity-theoretic sense).
- About lines 117-136, it was not clear to me what "precisions in inverse-cubic" means? Is that an additive error or a multiplicative error? So please clarify the hardness of approximation result.
- In lines 161-163, are you basically asking for an algorithm that matches your hardness result for approximating the capacity of MACs (see also page 24 and corresponding comment below)?
- Reference [20] does not seem to have a title, what it is about? If possible, please clarify.

-In the very end of page 24, I would appreciate a few more details on how the net construction basically matches your hardness result.

January 23, 2020

Responses to reviewer #3:

- *The submission is a collection of results, where I find the result (1) to be more interesting than the others. Results (2)-(3) are nice to have but basically follow immediately from the known literature on non-local games (once the connection of channel coding to non-local games is made). Result (4) is independent but may not be that groundbreaking. I would argue that the NP-hardness is not that surprising and moreover pretty straightforward to get from known complexity-theoretic results.*

We think that results (2)-(3) add substantial evidence to the complex nature of the entanglement-assisted capacity region of a MAC, and hence we chose to present results (1)-(4) on equal footing.

Regarding the reviewer's comment on result (4) not being that surprising, we would like to point out the following: In two widely popular (and now standard) textbooks on information theory, 'Elements of information theory' by Cover & Thomas and 'Network information theory' by El Gamal & Kim, the hardness of computing the MAC capacity region is either greatly underappreciated, or not addressed at all. We understood this as evidence that, prior to our work, the hardness of this problem was widely unknown or at least underappreciated, except for the specialists familiar with reference [19] and related papers. Therefore, we decided to emphasize result (4) in our paper, with the goal of introducing more colleagues from the classical information theory community to this interesting problem.

- *One might ask what the actual capacity is for general MACs with quantum assistance and this question is left open (same question for non-signalling assistance). However, this could very well be a hard question with no immediate answer and one might then also argue that this is where the follow-up works will come in.*

We agree with the reviewer's sentiment that deriving a formula for the entanglement-assisted capacity region of a general MAC is a hard and interesting problem. As pointed out in the Discussions section, this problem will be the subject of future work.

- *In line 22 (also line 120), the term "so-called single-letter formula that is in principle computable" might not be very clear to a general reader. Maybe one could refer to it as "an entropic optimisation problem of fixed, bounded dimension" or something along those lines.*

At the first occurrence of the term 'single-letter', we added a clarifying statement that 'single-letter formula' refers to an entropic optimization problem of fixed, bounded dimension, as suggested by the reviewer.

- *In lines 27-28 (also lines 146-151), I am not sure that I agree with "even unassisted classical MACs exhibit far more complex behavior than previously appreciated". After all, the MAC formula is an optimisation over product inputs which we know from optimisation theory to generally be hard. Also there are previous papers on the subject (e.g., [19]) and as far as I can tell it was actually appreciated that the computation of the MAC region can be hard (of course in the submission under review this is now made precise in some complexity-theoretic sense).*

Reflecting the prior work [19] on the evidence of the hardness of computing the MAC capacity region, we slightly weakened our statement in the second paragraph, replacing 'than previously appreciated' with 'than previously widely appreciated'. See also the above comment for more thoughts on this point.

- *About lines 117-136, it was not clear to me what "precisions in inverse-cubic" means? Is that an additive error or a multiplicative error? So please clarify the hardness of approximation result.*

We clarified the statement about the precision in the NP-hardness section to stress that we consider an additive error inverse-cubic in n .

- *In lines 161-163, are you basically asking for an algorithm that matches your hardness result for approximating the capacity of MACs (see also page 24 and corresponding comment below)?
In the very end of page 24, I would appreciate a few more details on how the net construction basically matches your hardness result.*

On the last page in the discussion of the net argument, we added a reference to the Exponential Time Hypothesis, and clarified that we do not actually claim optimality of the net argument. As stated in the text, the purpose of this section is merely to convince the reader that the 'naive' method of covering the set of product probability distributions with an epsilon-net is not far from optimal.

- *Reference [20] does not seem to have a title, what it is about? If possible, please clarify.*

Reference [20] is about unpublished work by J. Noetzel and A. Winter on a similar construction of MAC in terms of a non-local game, which we were made aware of in private communication with A. Winter. In the revised version of our manuscript, this communication is cited in Ref. [9], and part of the unpublished work by J. Noetzel and A. Winter appeared in a paper authored by J. Noetzel alone, Ref. [10].